# Standard-Deviation-Inspired Regularization for Improving Adversarial Robustness

**†Olukorede Fakorede**                                *olukoredefakorede@gmail.com, fakorede@iastate.edu*
*Department of Computer Science*
*Iowa State University, Ames, Iowa, USA*

**Modeste Atsague**                                                          *modeste@iastate.edu*
*Department of Computer Science*
*Iowa State University, Ames, Iowa,USA*

**Jin Tian**                                                               *jin.tian@mbzuai.ac.ae*
*Mohamed bin Zayed University of Artificial Intelligence*
*Abu Dhabi, United Arab Emirates*

**Reviewed on OpenReview:** *https://openreview.net/forum?id=6GfqNOCa1Y*

## Abstract

Adversarial Training ($AT$) has been demonstrated to improve the robustness of deep neural networks (DNNs) to adversarial attacks. $AT$ is a min-max optimization procedure wherein adversarial examples are generated to train a robust DNN. The inner maximization step of AT maximizes the losses of inputs w.r.t their actual classes. The outer minimization involves minimizing the losses on the adversarial examples obtained from the inner maximization. This work proposes a standard-deviation-inspired ($SDI$) regularization term for improving adversarial robustness and generalization. We argue that the inner maximization is akin to minimizing a modified standard deviation of a model's output probabilities. Moreover, we argue that maximizing the modified standard deviation measure may complement the outer minimization of the $AT$ framework. To corroborate our argument, we experimentally show that the $SDI$ measure may be utilized to craft adversarial examples. Furthermore, we show that combining the proposed $SDI$ regularization term with existing $AT$ variants improves the robustness of DNNs to stronger attacks (e.g., CW and Auto-attack) and improves robust generalization.

## 1 INTRODUCTION

The vulnerability of deep neural networks (DNNs) to adversarial perturbations is well documented in machine learning literature (Goodfellow et al., 2014; Moosavi-Dezfooli et al., 2016; Papernot et al., 2016; Szegedy et al., 2013), prompting concerns about the deployment of DNNs into safety-critical domains. Hence, for the safe deployment of DNNs, improving their robustness to adversarial perturbations is imperative.

Adversarial Training (AT) Goodfellow et al. (2014); Madry et al. (2018) has been demonstrated to be effective in improving the robustness of DNNs to adversarial attacks. $AT$ is a min-max optimization procedure, where the inner maximization step corresponds to finding adversarial examples in the direction of worst-case loss. The outer minimization minimizes the loss on the crafted adversarial examples. The success of $AT$ in improving the robustness of DNNs to adversarial perturbations has inspired a myriad of variants that have yielded better robustness or computational efficiency (Zhang et al., 2019; Wang et al., 2019; Li et al., 2019; Andriushchenko & Flammarion, 2020; Wong et al., 2020; Shafahi et al., 2019b). Furthermore, recent works have employed methods such as adversarial weight perturbation Wu et al. (2020), integration of hypersphere

---
†Corresponding author

embedding into $AT$ (Pang et al., 2020; Fakorede et al., 2023a), and loss re-weighting Zhang et al. (2020); Liu et al. (2021); Fakorede et al. (2023b); Zhang et al. (2023) to improve the performance of existing $AT$ variants.

In this work, we delve into a standard-deviation-inspired ($SDI$) measure proposed in a recent study in (Fakorede et al., 2024) for estimating the vulnerability of an input example. Drawing inspiration from the concept of standard deviation, which quantifies the dispersion of data points around the mean of a distribution, the $SDI$ measure aims to capture the dispersion of logits associated with incorrect classes in relation to the logits corresponding to the true class.

We draw a parallel between the inner maximization step of the $AT$ process and minimizing the $SDI$ loss function. Moreover, we argue that the outer minimization step of the $AT$ process, which seeks model parameters minimizing the loss on adversarial examples, is conceptually similar to maximizing the SDI measure. Both endeavors aim to enhance the likelihood of correctly classifying individual adversarial examples.

Unlike prevalent information-theoretic losses utilized in min-max $AT$ optimization, the $SDI$ measure operates independently from concepts like cross-entropy, entropy, or Kullback–Leibler divergence. Consequently, integrating the $SDI$ measure into existing $AT$ variants could complement information-theory-inspired losses and potentially enhance the performance of these variants. Therefore, we propose adding the $SDI$ loss as a regularization term to prominent $AT$ variants such as the *standard AT* (Madry et al., 2018) and *TRADES* (Zhang et al., 2019). The proposed regularization term is applied to outer minimization of the respective $AT$ variants to maximize the $SDI$ measure.

Our contributions are summarized as follows:

1. We propose utilizing the $SDI$ measure as a regularization term to existing $AT$ variants. Our extensive experiments show that our proposed approach further improves the robustness of existing $AT$ variants on strong attacks Auto attacks and CW attack and strong query-based black-box attack SPSA.

2. We experimentally show that the proposed $SDI$ regularization on existing $AT$ variants improves the generalization to other attacks not seen during adversarial training.

3. In addition, we establish a connection between minimizing the SDI measure and the inner maximization of the min-max $AT$ procedure. Specifically, we experimentally show that adversarial examples may be obtained from adversarial perturbations that minimize the $SDI$ metric. Furthermore, we compare the success rates of adversarial examples obtained using the $SDI$ metric with those obtained using cross-entropy loss and KL divergence on adversarial trained models.

## 2 BACKGROUND AND RELATED WORK

### 2.1 NOTATION

We use bold letters to represent vectors. We denote $\mathcal{D} = \{\mathbf{x}_i, y_i\}_{i=1}^n$ a data set of input feature vectors $\mathbf{x}_i \in \mathcal{X} \subseteq \mathbf{R}^d$ and labels $y_i \in \mathcal{Y}$, where $\mathcal{X}$ and $\mathcal{Y}$ represent a feature space and a label set, respectively.

Let $f_\theta : \mathcal{X} \to R^C$ denote a deep neural network (DNN) classifier with parameters $\theta$, and $|C|$ represents the number of output classes. For any $\mathbf{x} \in \mathcal{X}$, let the class label predicted by $f_\theta$ be $F_\theta(\mathbf{x}) = \arg\max_k f_\theta(\mathbf{x})_k$, where $f_\theta(\mathbf{x})_k$ denotes the $k$-th component of $f_\theta(\mathbf{x})$. $f_\theta(\mathbf{x})_y$ is the probability of $\mathbf{x}$ having label $y$.

We denote $\|\cdot\|_p$ as the $l_p$- norm over $\mathbf{R}^d$, that is, for a vector $\mathbf{x} \in \mathbf{R}^d, \|\mathbf{x}\|_p = (\sum_{i=1}^d |\mathbf{x}_i|^p)^{\frac{1}{p}}$. An $\epsilon$-neighborhood for $\mathbf{x}$ is defined as $B_\epsilon(\mathbf{x}) : \{\mathbf{x}' \in \mathcal{X} : \|\mathbf{x}' - \mathbf{x}\|_p \le \epsilon\}$. An adversarial example corresponding to a natural input $\mathbf{x}$ is denoted as $\mathbf{x}'$. We often refer to the loss resulting from the adversarial attack (inner maximization) as adversarial loss.

## 2.2 ADVERSARIAL ROBUSTNESS

Adversarial robustness is a machine learning model's capability to resist adversarial attacks. Over the past years, many methods (Guo et al., 2018; Buckman et al., 2018; Dhillon et al., 2018; Madry et al., 2018; Goodfellow et al., 2014; Zhang et al., 2019) have been proposed to improve adversarial robustness of neural networks. However, some of these defenses have been shown to provide a false sense of defense because they intentionally or inadvertently used obfuscated gradients in their defenses (Athalye et al., 2018).

In a seminal work, Madry et al. (2018) proposed Adversarial training (AT), which involves training the model with adversarial examples obtained under worst-case loss to improve robustness. Formally, $AT$ involves solving a min-max optimization as follows:

$$\min_{\boldsymbol{\theta}} \mathbb{E}_{(\mathbf{x},y)\sim\mathcal{D}} \left[ \max_{\mathbf{x}'\in B_\epsilon(\mathbf{x})} L(f_{\boldsymbol{\theta}}(\mathbf{x}'), y) \right] \tag{1}$$

where $L()$ represents the loss function, $y$ is the true label of input feature $\mathbf{x}$, and $\boldsymbol{\theta}$ are the model parameters. The inner maximization in Eq. (1) aims to obtain a worst-case adversarial version of the input $\mathbf{x}$ that increases the loss. The outer minimization then tries to find model parameters that would minimize this worst-case adversarial loss. The efficacy of $AT$ has spurred the development of numerous variants (Zhang et al., 2019; Wang et al., 2019; Wu et al., 2020; Pang et al., 2020).

A prominent variant $TRADES$ Zhang et al. (2019) proposed a principled regularization term that trades off adversarial robustness against natural accuracy. Wang et al. (2019) proposed $MART$, an $AT$ variant that differentiates between naturally misclassified examples that are used in the inner maximization of the $AT$ process, using this information to improve adversarial robustness. Wu et al. (2020) proposed adversarial weight perturbation, a double perturbation mechanism that employs the perturbation of inputs and weights to improve adversarial robustness. More recent $AT$ methods improve existing $AT$ variants by employing reweighting (Zhang et al., 2021; Liu et al., 2021; Fakorede et al., 2023b) or incorporating hypersphere embedding (Pang et al., 2020; Fakorede et al., 2023a).

The adversarial examples obtained in the inner maximization step of adversarial training methods are typically crafted using the projected gradient descent (PGD), maximizing the probability estimates of incorrect classes at the expense of the ground truth. Training on these specific adversarial types often leads to models performing well on the PGD adversarial attacks, on which the models are trained but generalizing relatively poorly to others. To address this, we propose a standard-deviation-inspired regularization term that explicitly maximizes the probability gap between incorrect classes and the ground truth while boosting the ground-truth probability. This proposed regularization operates directly on the model output logits, categorizing it as a form of logit regularization.

Most existing logit regularization variants (Mosbach et al., 2018; Kannan et al., 2018; Shafahi et al., 2019b; Summers & Dinneen, 2019; Kanai et al., 2021) involve utilizing techniques such as label smoothing and logit squeezing for improving adversarial robustness. These methods typically encourage smaller logit norms before softmax, which studies such as Shafahi et al. (2019b;a) associate with reduced overconfidence in predictions and improved adversarial robustness. However, the robustness achieved through these logit regularization methods has been criticized as potentially attributed to gradient obfuscation (Athalye et al., 2018; Engstrom et al., 2018; Lee et al., 2020; Raina et al., 2024). In contrast, our method operates on the post-softmax logits. It explicitly maximizes the probability gap between actual classes and the probability of incorrect classes, maximizing the confidence in the true classes of individual training samples. Our extensive experiments show the broad effectiveness of our approach in improving adversarial robustness to various adversarial attacks.

## 2.3 STANDARD DEVIATION AS A RISK MEASURE

The standard deviation measures the spread of a distribution around the mean of that distribution. The standard deviation of a distribution is given as:

$$SD = \sqrt{\frac{\sum_{i=1}^{N}(x_i - \mu)^2}{N-1}} \tag{2}$$

where $x_i$ is a data-point, $\mu$ is the population mean, and $N$ is the number of data-points in the distribution. A smaller *SD* value suggests that data points are more clustered, whereas a larger *SD* value indicates that data points are farther from the mean. The properties of standard deviation have made it a useful measure of risks in various domains. For example, the standard deviation is used as a risk measure in finance to measure market volatility and risk of assets and portfolios by indicating how much the returns of an asset deviate from the mean return (Artzner et al., 1999; Hull, 2012; Ross et al., 2019).

Drawing inspiration from the widely used standard deviation statistic, a recent work by Fakorede et al. (2024) proposes a modified standard deviation measure for scoring and characterizing the vulnerability of individual natural examples. Inspired by this work, our paper further argues a standard-deviation-inspired measure to be utilized to capture the risk of misclassification of training samples.

## 3 PROPOSED METHOD

Here, we justify the introduction of a Standard-Deviation-Inspired (*SDI*) measure as a regularization term into existing adversarial training approaches.

The *SDI* measure was originally proposed in Fakorede et al. (2024) for the purpose of estimating the vulnerability of natural examples. In this paper, we connect the *SDI* measure to the min-max optimization concept in adversarial training and use it as a regularization term.

### 3.1 THE *SDI* MEASURE

The *SDI* measure adopts an idea similar to standard deviation to characterize the spread of output probability vectors of DNN models for individual training examples. Specifically, the approach measures the variation of a model's estimated probabilities for incorrect classes around the model's estimated probability for the true label of individual input examples $\mathbf{x}$. Formally, given an input-label pair $(\mathbf{x}_i, y_i)$ and the output probabilities of a DNN model on input sample $\mathbf{x}_i$ denoted as $f_\theta(\mathbf{x}_i)$, the *SDI* measure is given as:

$$M_{SDI}(\mathbf{x}_i, y_i, \theta) = \sqrt{\frac{\sum_{k=1}^{|C|}(f_\theta(\mathbf{x}_i)_k - f_\theta(\mathbf{x}_i)_{y_i})^2}{|C| - 1}} \tag{3}$$

where $|C|$ is the number of output classes, $f_\theta(\mathbf{x}_i)_k$ is the model's estimated probability corresponding to class $k$, $f_\theta(\mathbf{x}_i)_{y_i}$ is the model's estimated probability of the true class, and $\theta$ denote the model parameters.

Under the condition where $f_\theta(\mathbf{x}_i)_{y_i} \geq \max_{k,k \neq y_i} f_\theta(\mathbf{x}_i)_k$, the $M_{SDI}(\mathbf{x}_i, y_i, \theta)$ measure serves to capture the vulnerability and risk of misclassification of individual examples $\mathbf{x}_i$. A smaller value of $M_{SDI}(\mathbf{x}_i, y_i, \theta)$ suggests that the output probabilities returned for sample $\mathbf{x}_i$ are more evenly distributed among classes, indicating a higher likelihood of misclassification as the model may misclassify it into any of the $k - 1$ incorrect classes.

### 3.2 AN *SDI*-ORIENTED PERSPECTIVE TO ADVERSARIAL TRAINING

In this section, we provide an explanation of the $M_{SDI}$ measure from the perspective of the min-max optimization framework of adversarial training.

*AT* methods are typically formulated as min-max optimization procedures. The inner maximization step of the *AT* approach involves generating adversarial examples $\mathbf{x}_i'$ from natural examples $\mathbf{x}_i$ by iteratively adjusting the input data in directions that maximize the loss, using projected gradient descent (PGD) algorithm as follows:

$$\mathbf{x}_i'^{t+1} \leftarrow \Pi_{\mathbf{x}_i' \in B_\epsilon(\mathbf{x}_i)}(x_i'^t + \alpha \cdot sign(\nabla_{\mathbf{x}_i'^t} L(\mathbf{x}_i'^t, y_i))). \tag{4}$$

where $\Pi$ is the projection operator and $L$ is a loss function.

Essentially, the adversarial examples produced during the inner maximization process are tailored to reduce the model's confidence in correctly classifying them into their true classes. Moreover, the resulting adversarial examples are untargeted, i.e., the inner maximization misclassifies the adversarial examples without prioritizing any particular incorrect class.

The $M_{SDI}(\mathbf{x}_i, y_i, \theta)$ measure estimates the vulnerability of individual inputs into a DNN model, using the spread of the model's estimated probabilities w.r.t. the model's estimated probability of the actual class of each input. Smaller values of $M_{SDI}(\mathbf{x}_i, y_i, \theta)$ for the output probability vector of a model indicate that the predicted probabilities are more concentrated or similar, reflecting lower confidence in the true class of the input. Therefore, the magnitude of $M_{SDI}(\mathbf{x}_i, y_i, \theta)$ value for an input-label pair $(\mathbf{x}_i, y_i)$ is indicative of the degree of risk in misclassifying $\mathbf{x}_i$. In contrast, a large value of $M_{SDI}(\mathbf{x}_i, y_i, \theta)$ indicates that the model assigns a high probability to class $y_i$ for $x_i$, suggesting strong confidence in the prediction and a low risk of misclassification.

This observation suggests that adversarial examples may be generated simply by finding adversarial perturbation along the gradient direction that minimizes the $M_{SDI}$ metric. We might use $M_{SDI}$ for generating adversarial examples as follows:

$$\mathbf{x}_i'^{t+1} \leftarrow \Pi_{\mathbf{x}' \in B_\epsilon(\mathbf{x}_i)}(x_i'^t - \alpha \cdot sign(\nabla_{\mathbf{x}_i'^t} M_{SDI}(\mathbf{x}_i'^t, y_i, \boldsymbol{\theta}))). \tag{5}$$

The above adversarial example generation is achieved using the widely adopted PGD algorithm (Madry et al., 2018), with the notable difference that the sign of the gradient is inverted to move in the opposite direction. For most $AT$ variants, adversarial examples in the inner maximization step are obtained by finding perturbations that maximize a cross-entropy loss function or a Kullback-Leibler divergence. The $SDI$ measure does not rely on information-theoretic measures. Therefore, it offers a complementary approach for finding adversarial examples. We provide experimental evidence for our claim in Sec. 4.5.

The outer minimization seeks model parameters that minimize the loss on the adversarial examples generated during the inner maximization step. Essentially, the outer minimization process aims to maximize the likelihood of correctly classifying individual adversarial training examples. Invariably, the outer minimization minimizes the likelihood of incorrect classification by increasing the probability gap between the example belonging to the label and belonging to incorrect classes. This conceptually aligns with the goal of maximizing the $SDI$ measure. Maximizing the $SDI$ metric encourages the model to correctly classify the input to its true class by widening the probability gap between the estimated probability for the true class and the estimated probabilities for other incorrect classes. Moreover, when $f_\theta(\mathbf{x}_i)_y \geq \max_{k,k \neq y} f_\theta(\mathbf{x}_i)_k$, maximizing the $M_{SDI}$ measure maximizes $f_\theta(\mathbf{x}_i)_y$.

### 3.3 SDI REGULARIZATION

Here, we propose the $SDI$ regularization term for improving adversarial training.

Typically, adversarial training techniques involve training models using adversarial examples generated by various forms of PGD attacks. However, this approach may lead to overly specialized models defending against PGD attacks, potentially causing poor generalization to different attack types. As discussed earlier, the $M_{SDI}$ metric introduced in the previous section has beneficial characteristics, particularly its ability to maximize the probability gap between the true class and the other classes. This property aligns well with the objectives of adversarial training, enhancing its effectiveness. Hence, to improve the robust generalization and performance of existing $AT$ methods, we propose adding a regularization term that maximizes the $M_{SDI}$ measure on each training example.

Maximizing the $M_{SDI}$ metric as a regularization term encourages the model to maximize the output probability of a training example belonging to its actual class, thus improving training. Moreover, since existing $AT$ variants depend on information-theoretic measures for both the inner maximization step and the outer minimization step, applying the $M_{SDI}$ metric as a regularization term offers a complementary addition to $AT$ methods that does not depend on the information-theoretic measures that these $AT$ methods are based. Lastly, maximizing the $M_{SDI}$ metric facilitates the widening of the probability gaps between the probability of the actual class of individual adversarial examples and the probabilities corresponding to incorrect classes, thus improving the discriminability of the model.

Note that maximizing the $M_{SDI}$ measure to improve $f_\theta(\mathbf{x}_i)_y$ is only valid when $f_\theta(\mathbf{x}_i)_y \geq \max_{k,k \neq y} f_\theta(\mathbf{x}_i)_k$. When $f_\theta(\mathbf{x}_i)_y < \max_{k,k \neq y} f_\theta(\mathbf{x}_i)_k$, maximizing $M_{SDI}$ may further minimize $f_\theta(\mathbf{x}_i)_y$, since the probability gap between each $f_\theta(\mathbf{x}_i)_{k,k \neq y}$ and $f_\theta(\mathbf{x}_i)_y$ is further increased to maximize the $M_{SDI}$ measure. Therefore,

we propose a regularization term $L_{SDI}$ that selectively maximizes the $M_{SDI}$ measure on samples whose output probabilities satisfies $f_\theta(\mathbf{x}_i)_y \geq \max_{k,k\neq y} f_\theta(\mathbf{x}_i)_k$.

We utilize the *multi-class margin* from (Koltchinskii & Panchenko, 2002) to determine input samples satisfying the desired conditions. For a DNN denoted by $f_\theta$ and the input-label pair $(\mathbf{x}_i, y_i)$, the margin $d_m(\mathbf{x}_i, y_i; \theta)$ is given as follows:

$$d_m(\mathbf{x}_i, y_i; \theta) = f_\theta(\mathbf{x}_i)_{y_i} - \max_{k,k\neq y_i} f_\theta(\mathbf{x}_i)_k \tag{6}$$

where $f_\theta(\mathbf{x}_i)_{y_i}$ is the model's predicted probability of the correct label $y_i$, and $\max_{k,k\neq y_i} f_\theta(\mathbf{x}_i)_k$ is the maximum prediction of the remaining classes.

The proposed *SDI* regularization term is formally described as follows:

$$L_{SDI}(\mathbf{x}_i, y_i; \theta) = \begin{cases} M_{SDI}(\mathbf{x}_i, y_i; \theta), & \text{if } d_m(\mathbf{x}_i, y_i; \theta) \geq 0 \\ 0, & \text{otherwise} \end{cases} \tag{7}$$

In this paper, we apply the $L_{SDI}(\mathbf{x}_i, y_i; \theta)$ regularization term to two prominent adversarial training methods: *standard AT* (Madry et al., 2018) and *TRADES (Zhang et al., 2019)*. We refer to the *SDI*-regularized *standard AT* and *TRADES* as *AT-SDI* and *TRADES-SDI* respectively. The regularized training objectives are stated as follows:
*AT-SDI*:

$$\sum_i L_{CE}(f_\theta(\mathbf{x}'_i), y_i) - \beta \cdot L_{SDI}(\mathbf{x}'_i, y_i, \theta) \tag{8}$$

*TRADES-SDI*:

$$\sum_i L_{CE}(f_\theta(\mathbf{x}_i), y) + \frac{1}{\lambda} \cdot KL(f_\theta(\mathbf{x}_i)\|f_\theta(\mathbf{x}'_i)) - \beta \cdot L_{SDI}(\mathbf{x}'_i, y_i, \theta) \tag{9}$$

where $\beta$ in Eq. (8) or (9) represents the regularization hyperparameter for controlling the weight of the *SDI* regularization term, and $KL$ in Eq. (9) represents Kullback–Leibler divergence. In the proposed *AT-SDI* and *TRADES-SDI*, the $L_{SDI}$ regularization term is selectively applied. The regularization term is only applied to adversarial training instances satisfying: $f_\theta(\mathbf{x}'_i)_y \geq \max_{k,k\neq y} f_\theta(\mathbf{x}'_i)_k$. If $f_\theta(\mathbf{x}'_i)_y < \max_{k,k\neq y} f_\theta(\mathbf{x}'_i)_k$ on a sample $\mathbf{x}'_i$, the normal *AT* or *TRADES* adversarial training is applied on $\mathbf{x}'_i$.

As an example, the proposed *AT-SDI* algorithm for adversarial training is presented in the following.

# 4 EXPERIMENTS

In this section, we conduct an extensive evaluation of the proposed method. To assess its versatility, we test it on various datasets, including CIFAR-10 (Krizhevsky et al., 2009), CIFAR-100 (Krizhevsky et al., 2009), SVHN (Netzer et al., 2011), and Tiny ImageNet Deng et al. (2009). We apply simple data augmentations, such as 4-pixel padding with $32 \times 32$ random crop and random horizontal flip, to each of the datasets. Additionally, we employ ResNet-18 (He et al., 2016) and WideResNet-34-10 (He et al., 2016) as the backbone models.

## 4.1 EXPERIMENTAL SETUP

### 4.1.1 Training Parameters.

We train the backbone networks using mini-batch gradient descent for 110 epochs, with a momentum of 0.9 and a batch size of 128. For training CIFAR-10, we used a weight decay of 5e-4, and for CIFAR-100, SVHN, and TinyImageNet, we used a weight decay of 3.5e-3. The initial learning rate was set to 0.1 (0.01 for CIFAR-100, SVHN, and TinyImageNet), and it was divided by 10 at the 75th epoch and then again at the 90th epoch.

---

**Algorithm 1** AT-SDI Algorithm.

**Input:** a neural network model with the parameters $\theta$, step size $\kappa$, $T$ PGD steps, a training dataset $\mathcal{D}$ of size $n$, $|C|$ is the number of classes, and hyperparameter $\beta$.

**Output:** a robust model with parameters $\theta^*$

1: **for** $epoch = 1$ to num\_epochs **do**
2:     **for** $batch = 1$ to num\_batchs **do**
3:         sample a mini-batch $\{(x_i, y_i)\}_{i=1}^{M}$ from $\mathcal{D}$;               ▷ mini-batch of size $M$.
4:         **for** $i = 1$ to M **do**
5:             $\mathbf{x}_i \leftarrow \mathbf{x}_i + 0.001 \cdot \mathcal{N}(0,1)$;     ▷ $\mathcal{N}(0, I)$ is a Gaussian distribution with zero mean and identity variance.
6:             **for** $t = 1$ to $T$ **do**
7:                 $\mathbf{x}_i' \leftarrow \Pi_{B_{\epsilon_i}(\mathbf{x}_i)}(x_i + \kappa \cdot sign(\nabla_{\mathbf{x}_i'} L_{CE}(f_\theta(\mathbf{x}_i'), y_i))$
8:             **end for**
9:         **end for**
10:         $M_{SDI}(\mathbf{x}_i', y_i; \theta) = \{\sum_{k=1}^{|C|} \frac{(f_\theta(\mathbf{x}_i')_k - f_\theta(\mathbf{x}_i')_{y_i})^2)}{|C|-1}\}^{0.5}$
11:         $d_m(\mathbf{x}_i', y_i; \theta) = f_\theta(\mathbf{x}_i')_{y_i} - \max_{k, k \neq y_i} f_\theta(\mathbf{x}_i')_k$
12:         **if** $d_m(\mathbf{x}_i', y_i; \theta) \geq 0$ **then**
13:             $L_{SDI}(\mathbf{x}_i', y_i; \theta) \leftarrow M_{SDI}(\mathbf{x}_i', y_i; \theta)$
14:         **else**
15:             $L_{SDI}(\mathbf{x}_i', y_i; \theta) \leftarrow 0$
16:         **end if**
17:         $\theta \leftarrow \theta - \eta \nabla_\theta \frac{1}{|M|} (\sum_{i=1}^{M} L_{CE}(f_\theta(\mathbf{x}_i'), y_i) - \beta \cdot L_{SDI}(\mathbf{x}_i', y_i; \theta))$
18:     **end for**
19: **end for**

---

### 4.1.2 Hyperparameters.

We set the value of $\beta$ to 3.0 for training AT-SDI and *TRADES-SDI* on CIFAR-10, SVHN, and TinyImagenet. For CIFAR-100 using AT-SDI and *TRADES-SDI*, we set $\beta$ to 3.0. When incorporating AWP (Wu et al., 2020) into *AT-SDI* and *TRADES-SDI*, we respectively set $\beta$ to 3.0 and 1.0. The hyperparameters are tuned using a validation set. We provide the sensitivity analysis of $\beta$ hyperparameter on *AT-SDI* and *TRADES-SDI* for CIFAR-10 using Wideresnet-34-10 in Tables 9 and 10.

### 4.2 BASELINES

We use prominent methods Standard *AT* (Madry et al., 2018) and *TRADES* (Zhang et al., 2019) as our baselines. In addition, we compare our results to other popular works *MART* (Wang et al., 2019), *AWP* (Wu et al., 2020), *MAIL* (Liu et al., 2021), and *ST-AT* (Li et al., 2023). All hyperparameters of the baseline methods remain consistent with those in their original papers. Nevertheless, we maintain consistency by using the same learning rate, batch size, and weight decay values as those utilized during the training of our proposed method.

### 4.3 THREAT MODELS

We evaluate the performance of the proposed method against strong attacks under *white-box* and *black-box* settings, as well as the Auto attack.

**White-box attacks.** These attacks have access to model parameters. To assess robustness on CIFAR-10 using Resnet-18 and Wideresnet-34-10, we employ the *PGD* attack with $\epsilon = 8/255$, step size $\kappa = 1/255$, and $K = 20$ iterations (PGD-20). Additionally, we utilize the *CW* attack (CW loss (Carlini & Wagner, 2017) optimized by *PGD-20*) with $\epsilon = 8/255$ and step size $1/255$. On SVHN and TinyImageNet, we use the *PGD* attack with $\epsilon = 8/255$, step size $\kappa = 1/255$, and $K = 20$ iterations.

**Black-box attacks.** In black-box settings, the adversarial attack method does not have access to the model parameters. We evaluate robust models trained on CIFAR-10 against strong black-box attack, SPSA (Uesato et al., 2018), with 100 iterations. These attacks use a perturbation size of 0.001 for gradient estimation, a learning rate of 0.01, and 256 samples for each gradient estimation. All black-box evaluations are conducted on trained Wideresnet-34-10.

**Auto attacks (AA).** Lastly, we assess the robustly trained models using *Autoattack* ($l_\infty$ and $l_2$) (Croce & Hein, 2020b), which is a powerful ensemble of attacks consisting of APGD-CE (Croce & Hein, 2020b), APGD-T (Croce & Hein, 2020b), FAB-T (Croce & Hein, 2020a), and Square (a black-box attack) (Andriushchenko et al., 2020).

## 4.4  PERFORMANCE EVALUATION

We present our experimental results and comparisons on various datasets using ResNet-18 and WideResNet-34-10 architectures. Specifically, results for CIFAR-10 on ResNet-18 and WideResNet-34-10 are summarized in Tables 1 and 2, respectively, while results for CIFAR-100, SVHN, and Tiny ImageNet using ResNet-18 are presented in Tables 3, 4, and 5, respectively. To further explore the versatility of the proposed method, we evaluate it using a lightweight backbone, VGG-16 architecture (Simonyan & Zisserman, 2014), on the CIFAR-10 dataset. The results are presented in Table 6.

Additionally, comparisons with other prominent baselines are provided in Table 7. Finally, we compare the perfomance of adversarial examples generated using the *SDI* metric approach described in Eq. (5) to adversarial examples crafted using cross-entropy and KL-divergence losses.

*The experiments were carried out three times using different random seeds. The mean and standard deviation were then calculated, with the results presented as mean ± std.*

### 4.4.1  Comparing *AT* and *TRADES* with their *SDI*-regularized variants.

In this comparison, we evaluate the performance of *AT* and *TRADES* against their respective variants with the *SDI* regularization term, *AT-SDI* and *TRADES-SDI*. Experimental findings indicate that the proposed regularization term enhances robustness against various adversarial attacks, including *Autoattacks* and *CW*. Specifically, when applied to ResNet-18 and WideResNet-34-10 architectures on CIFAR-10, *AT-SDI* demonstrates improvements over *AT* across all evaluated attacks (see Tables 1 and 2). For example, on WideResNet-34-10, *AT-SDI* outperforms *AT* in robustness against *PGD-20* (+0.45 %), *CW* (+2.54 %), and *Autoattacks* (+1.65 %). The improvement in robustness are achieved without a significant reduction in the natural accuracy.

Similarly, *TRADES-SDI* exhibits superior performance compared to *TRADES* on *PGD-20* (+1.19 %), *CW* (+2.06 %), and *Autoattacks* (+1.14%). Training with *TRADES-SDI* also exhibit a noticeable improvement of 0.67 % on the natural accuracy. Overall, *AT-SDI* achieves greater improvement in robustness against *CW* attacks compared to *TRADES-SDI*, while *TRADES-SDI* demonstrates better enhancement against *PGD-20* attacks compared to *AT* across ResNet-18 and WideResNet-34-10 architectures.

The proposed SDI regularization term also enhances robustness on CIFAR-100 when applied to ResNet-18 across all evaluated adversarial attacks (see Table (3)). The margin of improvement in robustness against adversarial attacks on CIFAR-100 appears to be larger than that observed on CIFAR-10 for both *AT-SDI* and *TRADES-SDI*. Similar improvements in robustness are observed when *AT-SDI* and *TRADES-SDI* are utilized to train Resnet-18 on SVHN dataset. Results in Table (4) show that *AT-SDI* outperforms *AT* on *CW* (+5.23%), *Autoattack* (+ 1.20%), and *PGD-20* (+2.43%).

Table 5 also clearly shows that the proposed training objective improves the robustness of Resnet-18 against all the evaluated attacks on Tiny Imagenet. The consistent improvement of performance across all the datasets tested validates the efficacy of the proposed *SDI* regularization term.

Finally, Table 6 demonstrates the effectiveness of the proposed training objectives on VGG-16 using the CIFAR-10 dataset. Incorporating the proposed $L_{SDI}$ regularization term into *AT* results in a marginal improvement in robustness to PGD-20, with a significant gain of 3.14% against CW and a 2.7% improvement against *Autoattack*. Similarly, *TRADES-SDI* surpasses *TRADES* with a 1.3% increase in robustness against *PGD-20*, a 1.71% gain against *CW*, and a 2.55% improvement against *Autoattack*.

Table 1: White-box attack robust accuracy for ResNet-18 on CIFAR-10.

| DEFENSE | NATURAL | PGD-20 | CW | AA |
|---|---|---|---|---|
| STANDARD-AT | $84.10_{\pm 0.09}$ | $52.78_{\pm 0.10}$ | $51.80_{\pm 0.14}$ | $47.95_{\pm 0.12}$ |
| **AT - SDI** (Ours) | $83.88_{\pm 0.10}$ | $\mathbf{53.43}_{\pm 0.06}$ | $\mathbf{53.71}_{\pm 0.09}$ | $\mathbf{49.56}_{\pm 0.07}$ |
| *TRADES* | $\mathbf{82.65}_{\pm 0.15}$ | $52.82_{\pm 0.13}$ | $51.82_{\pm 0.08}$ | $48.96_{\pm 0.11}$ |
| **TRADES - SDI** (Ours) | $82.04_{\pm 0.11}$ | $\mathbf{53.87}_{\pm 0.07}$ | $\mathbf{52.61}_{\pm 0.09}$ | $\mathbf{50.80}_{\pm 0.09}$ |

Table 2: White-box attack robust accuracy for Wideresnet-34-10 on CIFAR-10.

| DEFENSE | NATURAL | PGD-20 | CW | AA |
|---|---|---|---|---|
| *STANDARD AT* | $\mathbf{86.23}_{\pm 0.12}$ | $56.32_{\pm 0.10}$ | $54.95_{\pm 0.12}$ | $51.92_{\pm 0.09}$ |
| **AT-SDI** (Ours) | $86.11_{\pm 0.04}$ | $\mathbf{56.78}_{\pm 0.07}$ | $\mathbf{57.49}_{\pm 0.08}$ | $\mathbf{53.57}_{\pm 0.08}$ |
| *TRADES* | $84.70_{\pm 0.19}$ | $56.30_{\pm 0.16}$ | $54.51_{\pm 0.11}$ | $53.07_{\pm 0.13}$ |
| **TRADES-SDI** (Ours) | $\mathbf{85.37}_{\pm 0.11}$ | $\mathbf{57.49}_{\pm 0.16}$ | $\mathbf{56.57}_{\pm 0.11}$ | $\mathbf{54.21}_{\pm 0.07}$ |

Table 3: White-box attack robust accuracy for ResNet-18 on CIFAR-100.

| DEFENSE | NATURAL | $PGD-20$ | CW | AA |
|---|---|---|---|---|
| *STANDARD-AT* | $56.59_{\pm 0.22}$ | $28.18_{\pm 0.19}$ | $25.64_{\pm 0.18}$ | $24.07_{\pm 0.15}$ |
| **AT - SDI** (Ours) | $\mathbf{57.96}_{\pm 0.14}$ | $\mathbf{30.78}_{\pm 0.12}$ | $\mathbf{29.37}_{\pm 0.15}$ | $\mathbf{26.38}_{\pm 0.10}$ |
| *TRADES* | $56.96_{\pm 0.16}$ | $29.21_{\pm 0.12}$ | $25.57_{\pm 0.08}$ | $24.65_{\pm 0.06}$ |
| **TRADES - SDI** (Ours) | $\mathbf{60.68}_{\pm 0.11}$ | $\mathbf{31.21}_{\pm 0.04}$ | $\mathbf{28.73}_{\pm 0.04}$ | $\mathbf{26.45}_{\pm 0.04}$ |

Table 4: White-box attack robustness accuracy for ResNet-18 on SVHN.

| DEFENSE | NATURAL | $PGD-20$ | CW | AA |
|---|---|---|---|---|
| STANDARD-AT | $\mathbf{92.57}_{\pm 0.31}$ | $55.67_{\pm 0.14}$ | $52.92_{\pm 0.16}$ | $45.95_{\pm 0.12}$ |
| **AT-SDI** (Ours) | $92.10_{\pm 0.20}$ | $\mathbf{58.10}_{\pm 0.09}$ | $\mathbf{58.15}_{\pm 0.06}$ | $\mathbf{47.15}_{\pm 0.05}$ |
| *TRADES* | $\mathbf{90.83}_{\pm 0.14}$ | $57.27_{\pm 0.08}$ | $53.59_{\pm 0.05}$ | $46.45_{\pm 0.07}$ |
| **TRADES-SDI** (Ours) | $90.54_{\pm 0.07}$ | $\mathbf{59.21}_{\pm 0.10}$ | $\mathbf{56.39}_{\pm 0.10}$ | $\mathbf{49.21}_{\pm 0.03}$ |

Table 5: White-box attack robustness accuracy for ResNet-18 on Tiny Imagenet.

| DEFENSE | NATURAL | $PGD-20$ | CW | AA |
|---|---|---|---|---|
| *STANDARD-AT* | $48.83_{\pm 0.10}$ | $23.96_{\pm 0.05}$ | $21.85_{\pm 0.08}$ | $17.91_{\pm 0.09}$ |
| **AT-SDI** (Ours) | $\mathbf{49.73}_{\pm 0.04}$ | $\mathbf{24.79}_{\pm 0.03}$ | $\mathbf{23.16}_{\pm 0.06}$ | $\mathbf{20.01}_{\pm 0.08}$ |
| *TRADES* | $49.11_{\pm 0.18}$ | $22.82_{\pm 0.14}$ | $17.79_{\pm 0.16}$ | $16.82_{\pm 0.09}$ |
| **TRADES-SDI** (Ours) | $\mathbf{51.77}_{\pm 0.11}$ | $\mathbf{25.11}_{\pm 0.16}$ | $\mathbf{21.42}_{\pm 0.08}$ | $\mathbf{19.71}_{\pm 0.10}$ |

Table 6: White-box attack robust accuracy for VGG-16 on CIFAR-10.

| DEFENSE | NATURAL | $PGD-20$ | CW | AA |
|---|---|---|---|---|
| *STANDARD-AT* | $\mathbf{78.76}_{\pm 0.09}$ | $49.56_{\pm 0.04}$ | $46.98_{\pm 0.03}$ | $43.23_{\pm 0.05}$ |
| **AT - SDI** (Ours) | $78.69_{\pm 0.07}$ | $\mathbf{49.67}_{\pm 0.03}$ | $\mathbf{50.12}_{\pm 0.03}$ | $\mathbf{45.95}_{\pm 0.05}$ |
| *TRADES* | $\mathbf{80.42}_{\pm 0.10}$ | $48.78_{\pm 0.07}$ | $46.48_{\pm 0.09}$ | $43.96_{\pm 0.06}$ |
| **TRADES - SDI** (Ours) | $80.21_{\pm 0.11}$ | $\mathbf{50.08}_{\pm 0.09}$ | $\mathbf{48.19}_{\pm 0.06}$ | $\mathbf{46.51}_{\pm 0.08}$ |

### 4.4.2 Comparison with other prominent baselines.

Here, we compare our approach with other prominent and state-of-the-art methods from existing works, including *MART* (Wang et al., 2019), adversarial weight perturbation (*AWP*) (Wu et al., 2020), ST-AT (Li et al., 2023), *LAS AT* (Jia et al., 2022), *LOAT* (Yin & Ruan, 2024) and *Randomize-AT* (Jin et al., 2023). Additionally, for a fair comparison with *AWP*, we combine *AT-SDI* and *TRADES-SDI* with *AWP* and denote them as *AT-SDI + AWP* and *TRADES-SDI + AWP*, respectively. In both *AT-SDI + AWP* and *TRADES-SDI + AWP*, the *SDI* regularization term is employed for perturbing the network weights.

Experimental results displayed in Table 7 show that *AT-SDI* outperforms all existing baselines in robustness against *CW* attacks. Additionally, *AT-SDI* outperforms *TRADES* and *MART* against *Autoattacks*. However, *AT-SDI* slightly underperforms compared to *MART* against *PGD-20*. Furthermore, *AT + AWP* also marginally outperforms *AT-SDI* against Autoattacks. On the other hand, *TRADES-SDI* achieves better performance than all baselines on *CW* and *Autoattacks*.

When compared to recent state-of-the-art methods, *AT-SDI* and *TRADES-SDI* demonstrate superior performance on *CW* and *SPSA* attacks. *TRADES-SDI* outperforms *LAS AT* on *CW* (+0.82%), *Auto-attacks* (+0.66%), and *SPSA* (+ 1.1%). Although *LOAT* moderately performs better than *AT-SDI* and *TRADES-SDI* on *PGD-20*, both methods show significantly better than *LOAT* against *CW*, *AA*, and *SPSA*. While *Randomize AT* and *CAT* slightly surpass *AT-SDI* and *TRADES-SDI* on *PGD-20*, *AT-SDI* and *TRADES-SDI* perform better on *CW* and *SPSA* attacks. Moreover, *Randomize AT* and *CAT* take roughly twice as long to train, making *AT-SDI* and *TRADES-SDI* significantly more efficient.

Combining our approach with *AWP* further improves robustness against the evaluated attacks. Specifically, *AT-SDI + AWP* and *TRADES-SDI + AWP* outperform *AWP + AT* and *TRADES + AWP* against every adversarial attack. *AT-SDI + AWP* and *TRADES-SDI + AWP* demonstrate improved performance across all attacks. *AT-SDI + AWP* enhances robustness to *PGD-20*, *CW* attacks, and Autoattacks over *AWP* by 2.11%, 4.38%, and 2.79%, respectively. *AT-SDI+AWP* also considerably outperforms *AWP* robustness against *SPSA*, a strong query-based blackbox attack, by 2.99%. Additionally, *AT-SDI + AWP* achieves superior performance on natural samples. *TRADES-SDI + AWP* improves performance over *AWP* on all the attacks but dips by 0.15% in performance on natural examples.

Table 7: Comparison with other state-of-the-art baselines under white-box and black-box attacks on CIFAR-10 for Wideresnet-34-10.

| Defense | NATURAL | PGD-20 | CW | AA | SPSA |
|---|---|---|---|---|---|
| *Standard AT* | $86.23_{\pm 0.12}$ | $56.32_{\pm 0.10}$ | $54.95_{\pm 0.12}$ | $51.92_{\pm 0.09}$ | $61.05_{\pm 0.05}$ |
| *TRADES* | $84.70_{\pm 0.19}$ | $56.30_{\pm 0.16}$ | $54.51_{\pm 0.11}$ | $53.07_{\pm 0.13}$ | $61.15_{\pm 0.08}$ |
| *MART* | $84.17_{\pm 0.05}$ | $58.10_{\pm 0.15}$ | $54.51_{\pm 0.09}$ | $51.11_{\pm 0.04}$ | $58.91_{\pm 0.06}$ |
| *MAIL* ((Liu et al., 2021)) | $86.81_{\pm 0.11}$ | $60.49_{\pm 0.13}$ | $51.45_{\pm 0.11}$ | $47.11_{\pm 0.13}$ | $59.25_{\pm 0.07}$ |
| *ST-AT* ((Li et al., 2023)) | $84.91_{\pm 0.09}$ | $57.52_{\pm 0.07}$ | $55.11_{\pm 0.08}$ | $53.54_{\pm 0.08}$ | $61.34_{\pm 0.07}$ |
| *LAS-AT* ((Jia et al., 2022)) | $86.23_{\pm 0.13}$ | $56.50_{\pm 0.11}$ | $55.75_{\pm 0.13}$ | $53.55_{\pm 0.09}$ | $61.21_{\pm 0.10}$ |
| *Randomize-AT* Jin et al. (2023)* | $85.99_{\pm 0.12}$ | $58.41_{\pm 0.16}$ | $56.14_{\pm 0.14}$ | $54.15_{\pm 0.11}$ | $61.59_{\pm 0.07}$ |
| *CAT* Liu et al. (2023)* | $86.24_{\pm 0.17}$ | $57.51_{\pm 0.14}$ | $55.93_{\pm 0.13}$ | $54.13_{\pm 0.16}$ | $61.37_{\pm 0.10}$ |
| *LOAT* ((Yin & Ruan, 2024)) | $84.17_{\pm 0.19}$ | $58.67_{\pm 0.12}$ | $55.70_{\pm 0.09}$ | $52.35_{\pm 0.08}$ | $60.27_{\pm 0.10}$ |
| **AT-SDI** (ours) | $86.11_{\pm 0.04}$ | $56.78_{\pm 0.07}$ | $57.49_{\pm 0.08}$ | $53.57_{\pm 0.08}$ | $62.46_{\pm 0.05}$ |
| **TRADES-SDI** (ours) | $85.37_{\pm 0.11}$ | $57.49_{\pm 0.16}$ | $56.57_{\pm 0.11}$ | $54.21_{\pm 0.07}$ | $62.31_{\pm 0.06}$ |
| **AT-SDI + AWP** (ours) | $\mathbf{88.21}_{\pm 0.06}$ | $60.15_{\pm 0.05}$ | $\mathbf{60.30}_{\pm 0.05}$ | $56.71_{\pm 0.06}$ | $\mathbf{65.56}_{\pm 0.04}$ |
| **TRADES-SDI + AWP** (ours) | $85.21_{\pm 0.12}$ | $\mathbf{60.72}_{\pm 0.08}$ | $58.15_{\pm 0.07}$ | $\mathbf{56.82}_{\pm 0.05}$ | $63.41_{\pm 0.05}$ |

Table 8: Comparison with other baselines under white-box and black-box attacks on Tiny Imagenet for ResNet-18.

| Defense | NATURAL | PGD-20 | CW | AA | SPSA |
|---|---|---|---|---|---|
| *Standard AT* | $48.83_{\pm 0.14}$ | $23.96_{\pm 0.11}$ | $21.85_{\pm 0.11}$ | $17.91_{\pm 0.12}$ | $26.93_{\pm 0.10}$ |
| *TRADES* | $49.11_{\pm 0.21}$ | $22.82_{\pm 0.18}$ | $17.79_{\pm 0.23}$ | $16.82_{\pm 0.20}$ | $27.41_{\pm 0.15}$ |
| *MART* | $46.01_{\pm 0.07}$ | $26.03_{\pm 0.11}$ | $22.08_{\pm 0.17}$ | $19.18_{\pm 0.09}$ | $28.15_{\pm 0.08}$ |
| *MAIL* ((Liu et al., 2021)) | $49.72_{\pm 0.31}$ | $24.09_{\pm 0.29}$ | $21.21_{\pm 0.23}$ | $17.42_{\pm 0.19}$ | $26.68_{\pm 0.15}$ |
| *ST-AT* ((Li et al., 2023)) | $48.61_{\pm 0.08}$ | $23.85_{\pm 0.11}$ | $18.43_{\pm 0.10}$ | $17.29_{\pm 0.08}$ | $27.91_{\pm 0.08}$ |
| *AWP* ((Wu et al., 2020)) | $48.89_{\pm 0.09}$ | $24.97_{\pm 0.13}$ | $22.39_{\pm 0.11}$ | $18.68_{\pm 0.17}$ | $28.23_{\pm 0.10}$ |
| **AT-SDI** (ours) | $49.73_{\pm 0.10}$ | $24.79_{\pm 0.08}$ | $23.16_{\pm 0.14}$ | $20.01_{\pm 0.06}$ | $28.95_{\pm 0.09}$ |
| **TRADES-SDI** (ours) | $51.77_{\pm 0.24}$ | $25.11_{\pm 0.17}$ | $21.42_{\pm 0.14}$ | $19.71_{\pm 0.12}$ | $28.36_{\pm 0.10}$ |
| **AT-SDI + AWP** (ours) | $50.12_{\pm 0.27}$ | $\mathbf{26.14}_{\pm 0.16}$ | $\mathbf{24.27}_{\pm 0.13}$ | $\mathbf{20.47}_{\pm 0.11}$ | $\mathbf{29.07}_{\pm 0.12}$ |
| **TRADES-SDI + AWP** (ours) | $\mathbf{52.87}_{\pm 0.28}$ | $25.56_{\pm 0.17}$ | $23.59_{\pm 0.14}$ | $19.83_{\pm 0.15}$ | $28.41_{\pm 0.11}$ |

*It takes approximately twice as much time to train compared to our methods.

### 4.4.3 SDI-regularization Improves Generalization of Adversarial Training.

Most $AT$ methods involve training with a specific type of adversarial examples crafted by maximizing either the cross-entropy or KL-divergence measure using $PGD$. Therefore, the adversarial examples utilized for adversarial training do not entirely reflect the universe of all possible adversarial attacks that a robust model may encounter. This limitation can lead to poor generalization of adversarially trained models to other types of adversarial examples (Song et al., 2018).

Typically, adversarial training methods exhibit significantly higher performance on $PGD$ attacks, as evident from the experimental results tables. However, when subjected to other types of attacks, the performance of robust models tends to diminish. For example, it can be observed from the tables that the robust accuracy on $CW$ and $AA$ attacks are notably lower compared to $PGD$-$20$.

The introduction of the $L_{SDI}$ regularization term to the standard $AT$ and $TRADES$ improves their performances against other attacks. Unlike other $AT$ methods, $AT$-$SDI$ considerably improves the robustness against $CW$ and $AA$ on all the datasets evaluated. In fact, training a model using the proposed $AT$-$SDI$ consistently improves the performance of the resulting robust model to $CW$ attack and achieve better performance over $PGD$-$20$ on $CIFAR$-$10$ dataset, as may be observed in Tables 1, 2, and 7. Note that $CW$ adversarial examples are not used for training, yet better robust accuracies are recorded compared to $PGD$ adversarial examples, which are typically used for adversarial training. Significant improvement in robustness against $CW$ and $AA$ can also be observed on other datasets, CIFAR-100, SVHN, and Tiny Imagenet. The SDI regularization term also improves the performance of $TRADES$ on $CW$ and $AA$ attacks. Experiments in Tables 1 - 5 show that the SDI regularization reduces the performance gap between PGD-20 and the other attacks.

Combining $AWP$ (Wu et al., 2020) with $AT$-$SDI$ achieves a high robustness of 60.30% on CW, improving $AWP$ by 4.38% on CIFAR-10. Further, the robustness performance on $CW$ is better than the robustness performance on $PGD$-$20$. Also in Table 7, the improvement in performance is noticeable against $AA$ and $SPSA$ attacks. Similarly, the improvements in robustness to $CW$ $AA$ can be observed in Table 7, when the $L_{SDI}$ regularization is applied to $TRADES + AWP$.

Overall, the proposed $L_{SDI}$ regularization term consistently minimizes the performance gaps between robustness to $PGD$-$20$ adversarial examples and other types of adversarial examples. This supports our argument that the $L_{SDI}$ regularization improves the generalization of adversarial training. An intuitive explanation for this observation is that the $L_{SDI}$ regularization is not dependent on the specific algorithmic nuances of individual adversarial attacks and defenses. Instead, it explicitly maximizes the probability gaps between the probability of the true class of each adversarial example and the probabilities corresponding to incorrect classes.

### 4.4.4 Sensistivity Analysis of the hyper-parameter $\beta$

Here, we study the influence of the regularization hyper-parameter $\beta$ on $AT$ - $SDI$ and $TRADES$-$SDI$ performance.

We trained WideResNet-34-10 using $AT$-$SDI$ with $\beta$ values of 1.0, 2.0, 3.0, 4.0, and 5.0, and $TRADES$-$SDI$ with $\beta$ values of 1.0, 2.0, 2.5, 3.0, 4.0, and 5.0. We present the results in Tables 9 and 10, which shows that increasing the value of $\beta$ leads to moderate reduction in the natural accuracy of $AT$-$SDI$ and $TRADES$-$SDI$. The robust accuracy of $PGD$-$20$ remains relatively stable for various values of $\beta$ on $AT$-$SDI$ and $TRADES$-$SDI$. Nevertheless, $AT$-$SDI$ exhibits noticeable improvement in robustness against $CW$ attack as $\beta$ increases. The performance of $AT$-$SDI$ against $Autoattack$ also improves with increasing $\beta$ value but diminishes as $\beta$ gets too large. We selected $\beta = 3.0$ since it maintains a good balance between the natural and robust accuracy.

Like $AT$-$SDI$, $TRADES$-$SDI$ is also sensitive to $\beta$. The variations in natural accuracy are moderate when $\beta$ is varied. The robustness to $PGD$-$20$ remains relatively stable as $\beta$ varies. It can be observed from 10 that the robust performance on $CW$ attack and $AA$ increases as $\beta$ increases, but both decrease slightly when $\beta$ is set to 5.0.

Table 9: Sensitivity analysis on $\beta$ on AT-SDI for Wideresnet-34-10 on CIFAR-10.

| $\beta$ | NATURAL | PGD-20 | CW | AA |
|---|---|---|---|---|
| 1.0 | $86.75_{\pm0.05}$ | $56.77_{\pm0.07}$ | $57.07_{\pm0.10}$ | $53.16_{\pm0.06}$ |
| 2.0 | $86.52_{\pm0.07}$ | $56.84_{\pm0.07}$ | $57.19_{\pm0.09}$ | $53.44_{\pm0.09}$ |
| 3.0 | $86.11_{\pm0.04}$ | $56.78_{\pm0.07}$ | $57.49_{\pm0.08}$ | $53.57_{\pm0.08}$ |
| 4.0 | $85.78_{\pm0.11}$ | $56.69_{\pm0.10}$ | $57.49_{\pm0.06}$ | $53.39_{\pm0.09}$ |
| 5.0 | $85.59_{\pm0.10}$ | $56.37_{\pm0.09}$ | $57.81_{\pm0.08}$ | $52.73_{\pm0.11}$ |

Table 10: Sensitivity analysis on $\beta$ on *TRADES-SDI* for Wideresnet-34-10 on CIFAR-10.

| $\beta$ | NATURAL | PGD-20 | CW | AA |
|---|---|---|---|---|
| 1.0 | $85.51_{\pm0.15}$ | $57.27_{\pm0.18}$ | $55.66_{\pm0.09}$ | $53.43_{\pm0.10}$ |
| 2.0 | $85.43_{\pm0.10}$ | $57.29_{\pm0.14}$ | $56.01_{\pm0.08}$ | $53.61_{\pm0.7}$ |
| 2.5 | $85.39_{\pm0.12}$ | $57.36_{\pm0.13}$ | $56.28_{\pm0.11}$ | $53.88_{\pm0.09}$ |
| 3.0 | $85.37_{\pm0.11}$ | $57.49_{\pm0.16}$ | $56.57_{\pm0.11}$ | $54.21_{\pm0.07}$ |
| 4.0 | $85.24_{\pm0.13}$ | $57.33_{\pm0.15}$ | $56.66_{\pm0.13}$ | $54.11_{\pm0.09}$ |
| 5.0 | $85.21_{\pm0.11}$ | $57.31_{\pm0.14}$ | $56.59_{\pm0.12}$ | $54.05_{\pm0.11}$ |

### 4.4.5 Computational Cost

We conducted all experiments using a single core of an AMD EPYC 7513 processor, an Nvidia A100 SXM4 80 GB GPU, and 128 GB of RAM. When the proposed $L_{SDI}$ regularization term is added to $AT$, it increases the training time per epoch by no more than 4 seconds for ResNet-18. For context, popular regularization losses like KL-divergence and mean square error add up to 8 seconds and 10 seconds per epoch, respectively, under similar conditions and computational resources. Therefore, the $L_{SDI}$ regularization is lightweight and introduces minimal overhead compared to KL divergence and mean square error losses.

---

**Algorithm 2** SDI-PGD Algorithm.

---

    **Input:** a neural network model with the parameters $\theta$, step size $\kappa$, natural examples $\mathbf{x}_i$ in a labelled dataset $\mathcal{D}$ of size $n$ and $|C|$ is the number of classes.

    **Output:** Adversarial examples $\mathbf{x}_i'$

1: Sample $(\mathbf{x}_i, y_i)$ from $\mathcal{D}$;
2: $\mathbf{x}_i' \leftarrow \mathbf{x}_i + 0.001 \cdot \mathcal{N}(0,1)$;                 ▷ $\mathcal{N}(0,I)$ is a Gaussian distribution with zero mean and identity variance.
3: **for** $t = 1$ to $T$ **do**                                            ▷ T is the number of PGD iteration steps.
4:      $M_{SDI}(\mathbf{x}_i', y_i; \theta) = \{\sum_{k=1}^{|C|} \frac{(f_\theta(\mathbf{x}_i')_k - f_\theta(\mathbf{x}_i')_{y_i})^2)}{|C|-1}\}^{0.5}$
5:      $\mathbf{x}_i' \leftarrow \Pi_{B_\epsilon(\mathbf{x}_i)}(x_i - \kappa \cdot sign(\nabla_{\mathbf{x}_i'} M_{SDI}(\mathbf{x}_i', y_i; \theta))$             ▷ Π denotes the projection operator.
6: **end for**
7: **return** $\mathbf{x}_i'$                                                    ▷ Return adversarial example.

---

### 4.5 ADVERSARIAL ATTACK USING THE *SDI* METRIC

In Section 3, we argue that the proposed *SDI* metric in Eq. (3) can be utilized for crafting adversarial examples.

Here, we compare PGD-based adversarial examples optimized using the *SDI* metric with existing popular PGD-based adversarial examples crafted using information-theoretic measures such as cross-entropy (Madry et al., 2018) and KL-divergence (Zhang et al., 2019). The $M_{SDI}$-measure-optimized adversarial examples are crafted using the approach described in Eq. (5). We compare the performances of each approach under $AT$ and *TRADES*. In each case, the attack is obtained using the conventional attack settings: 20 PGD iterations, perturbation bound $\epsilon$ 0.031, and step size 0.003. The algorithm for obtaining adversarial examples using the $M_{SDI}$ measure is provided in Algorithm 2.

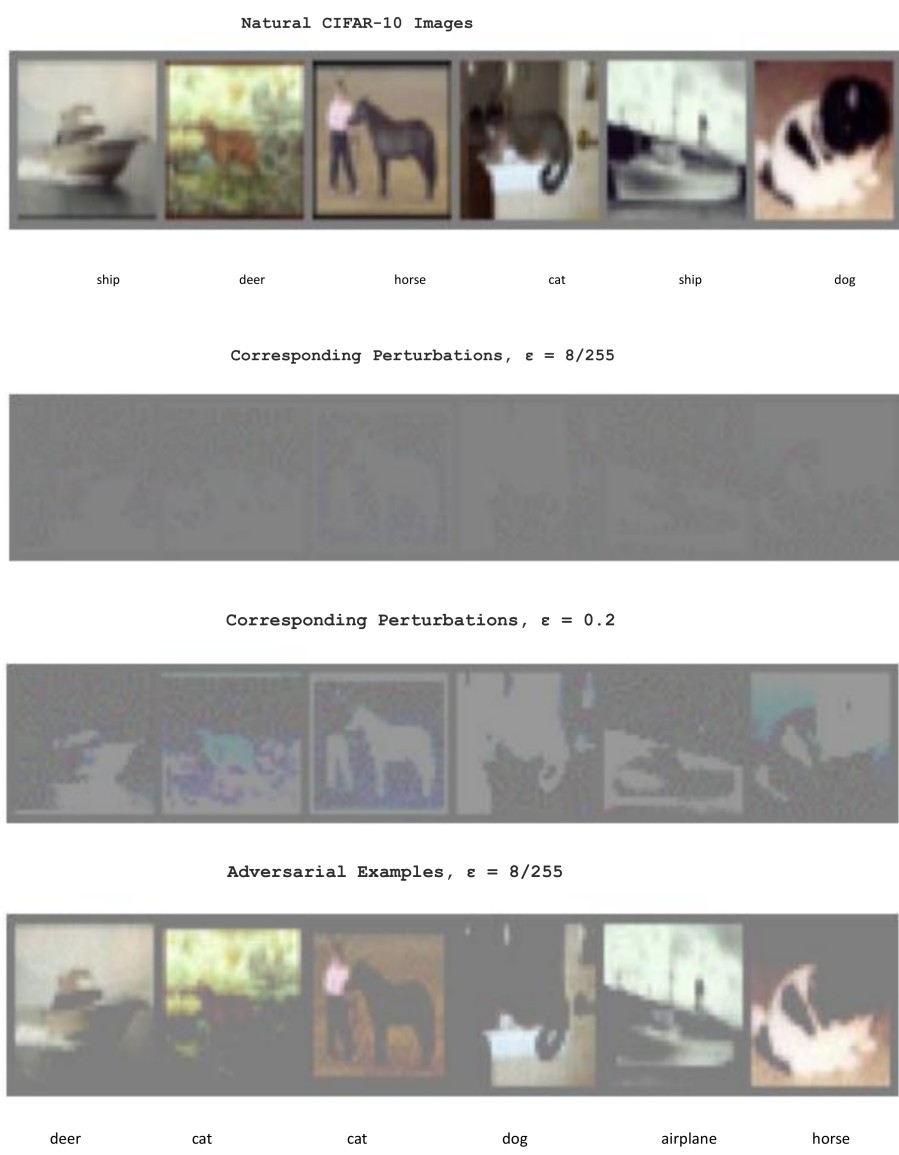

Figure 1: Comparison of natural CIFAR-10 images with the adversarial perturbations and adversarial examples obtained by SDI-PGD attack defined in Eq. (5). Images in the first row represent natural CIFAR-10 images and their correct labels. The second and third rows represent the corresponding $l_\infty$ adversarial perturbations, with $\epsilon = 8/255$ and 0.2 respectively. The fourth row represents the corresponding adversarial examples and their incorrect labels for each image in row one.

We display some adversarial examples obtained using the proposed $M_{SDI}$-measure-optimized approach and their corresponding misclassified labels in the last row of Figure 1. Additionally, we display the corresponding adversarial perturbations of the original images in rows two and three.

We compare the robustness of two prominent adversarial training variants, *AT* and *TRADES*, against *PGD*-based adversarial examples generated using cross-entropy loss, KL-divergence, and the SDI measure. The experimental results are presented in Table 11. It is evident from the table that adversarial examples crafted using the SDI metric exhibit significantly greater strength compared to those crafted using KL-divergence.

Nevertheless, adversarial examples generated with the cross-entropy loss demonstrate a marginally greater strength than those obtained using the proposed SDI metric.

The results show that the *SDI* metric produces 'useful gradients' for generating adversarial examples, a desirable quality as discussed in prior studies (Athalye et al., 2018; Papernot et al., 2017). This suggests that optimizing the *SDI* metric for improving adversarial robustness would not lead to gradient obfuscation (Athalye et al., 2018).

Table 11: Comparison of successes of PGD attacks crafted using cross-entropy, KL-divergence, and *SDI* measure on defences *AT* (Madry et al., 2018) and *TRADES* (Zhang et al., 2019) on CIFAR-10 for Resnet-18.

| PGD ATTACK MEASURE | $AT$ | $TRADES$ |
|---|---|---|
| CROSS-ENTROPY | $\mathbf{52.78}_{\pm 0.10}$ | $\mathbf{52.82}_{\pm 0.12}$ |
| KL-DIVERGENCE | $68.03_{\pm 0.15}$ | $68.87_{\pm 0.14}$ |
| **SDI** | $53.95_{\pm 0.14}$ | $54.32_{\pm 0.19}$ |

## 5 CONCLUSION

We introduce a novel regularization term based on a standard deviation-inspired (*SDI*) measure to improve adversarial robustness. The *SDI* measure captures the spread of a model's estimated probabilities with respect to the true class of each input. We establish a connection between optimizing the *SDI* measure and the min-max optimization procedure in adversarial training. Specifically, we illustrate that the SDI measure may be optimized for generating adversarial examples by seeking perturbations that minimize the SDI measure.

We demonstrate with experimental study that maximizing the SDI measure on adversarial training examples contributes to improving the robustness of existing adversarial training methods. Empirical results indicate that the proposed regularization significantly improves existing adversarial training variants' robustness and generalization capabilities.

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
