# OpenReview forum: "Standard-Deviation-Inspired Regularization for Improving Adversarial Robustness"
_TMLR — Accepted by TMLR_

### Review · Reviewer_d1mo · 2024-09-25

**Summary Of Contributions:**

The paper introduces a regularisation term to the previous “Adversarial Training (AT)” methods to improve robustness. The regulariser is based on a constrained standard deviation that measures the dispersion of the probabilities assigned to the incorrect class with the probability of the correct class.  AT methods are usually a bi-level min-max optimisation and the proposed regulariser is only applied to the outer optimisation.

**Audience:**

Yes

**Claims And Evidence:**

No

**Requested Changes:**

Please make changes according to my previous comments.

**Strengths And Weaknesses:**

**Strengths**:

S1. The paper is mostly very well-written and structured. It is coherent and straightforward to follow and understand.

S2. The proposed regulariser is easy to interpret and has a natural meaning for adversarial training.

**Weaknesses**:

W1. In the abstract, there is a sentence explaining that AT is un-targeted. This doesn’t seem particularly important for the paper to be pointed out in the abstract. I think a mention in the introduction would suffice.

W2. In the contributions: “we argue … “ is not a proper contribution. The contribution is the second part, which is “experimentally show …”.

W3. The paper doesn’t have a clear and organised literature review. It’s partly in the introduction and Section 2 which only mentions AT works. More importantly, the purpose of a literature review is to identify the research gaps to better motivate and position your work. At the very least, the authors should point out that the current AT methods still need improvement.

W4. In Section 2.3, the standard deviation is a basic and known statistic, and in my opinion, it doesn’t need two paragraphs with examples to explain.  Instead, it is better to add more details about Fakorede et al. 2024, since your paper is built on that.

W5. In section 3.2, the fourth paragraph: “this observation suggests that …” is not a correct conclusion. I argue the contrary, i.e., minimizing SDI can yield one set of adversarial examples. But, it doesn’t necessarily mean that AT generates adversarial examples in the direction of minimizing SDI.  It might be the case under the $f_{\theta}(x_i)y > max_{k,k\neq y}f_{\theta}(x_i)_k$ constraint, but that needs to be proven.

W6. In section 3.3, the second paragraph claims that the introduced $M_{SDI}$ metric possesses favourable attributes. However, these attributes are not identified or explained anywhere in the paper. There are mentions that SDI is not an information-theoretic measure like cross-entropy, but this alone is not a useful feature of SDI.

W7. In Algorithm 1, add \beta to the Input arguments.

W8. In section 4.1.2, explain the model selection procedure. Since \beta trade-offs accuracy and robustness, which one is the priority for the hyperparameter tuning? For example, in section 4.4.4 you say “We selected $\beta=3$ since it maintains a good trade-off between natural and robust accuracy”. How do you define “goodness”?

W9. Tables 7 and 8 captions say ablation studies while they’re not actually ablations. Sensitivity analysis makes more sense.

W10. In section 4.5, move Algorithm 2 and the relevant descriptions to Section 3. Also, can you elaborate on the last paragraph? What does ‘useful gradients’ mean, and how does SDI optimisation prevent gradient obfuscation?

W11. In general, the results show that the introduced regulariser (Tables 1 to 5) can only marginally improve the robustness in terms of accuracy (by less than 1% or 2%).  Could the authors think of any other aspects that SDI can help with? Or maybe some AT methods than SDI can significantly improve?

---

### Review · Reviewer_7832 · 2024-09-25

**Summary Of Contributions:**

This paper proposes using standard-deviation-inspired (SDI) regularization term to improve adversarial robustness and generalization in deep neural networks. The main contributions are:
	1. Proposal of the SDI measure as a regularization term, which quantifies the dispersion of a model's output probabilities relative to the true class probability. For an input-label pair $(x\_i, y\_i)$, the SDI measure is defined as:
$M\_{SDI}(x\_i, y\_i, θ) = \sqrt{\Sigma\_{k=1}^{|C|} (f\_θ(x\_i)\_k - f\_θ(x\_i){y\_i})^2) / (|C| - 1)}$
where $f\_θ(x\_i)\_k$ is the model's estimated probability for class $k$, and $f\_θ(x\_i)_{y\_i}$ is the probability for the true class.
	2. Argumentation that minimizing SDI is conceptually similar to the inner maximization step in adversarial training, while maximizing SDI aligns with the outer minimization step.
	3. Demonstration that the SDI measure can be used to generate adversarial examples. For example, the AT-SDI objective is formulated as: $\Sigma\_i L\_{CE}(f\_θ(x'\_i), y\_i) - \beta \cdot L\_{SDI}(x'\_i, y\_i, \theta)$
where L_CE is the cross-entropy loss, x'_i are adversarial examples, and β is a hyperparameter controlling the strength of the regularization.
	4. Empirical evidence showing that incorporating the SDI regularization term into existing adversarial training methods (AT and TRADES) improves robustness against attacks like CW and Auto-attack.
   5. Demonstration of improved robust generalization, particularly for attacks not seen during training.

**Audience:**

Yes

**Broader Impact Concerns:**

The paper doesn't explicitly discuss broader impact concerns. A broader impact statement could be added discussing:

Potential positive impacts: Improved adversarial robustness could lead to more reliable and secure AI systems in critical applications.

Potential negative impacts: This technique could potentially be used to create more robust adversarial examples.

**Claims And Evidence:**

Yes

**Requested Changes:**

Critical:

	1. Does the argument on alignment of SDI measure with minimization and maximization hold only when the maximum probability class is the true class? Please clarify.

	2. Two of the main claims seems to be slightly contradictory: (1) you argue that SDI measure minimization and maximization aligns with minimization and maximization of AT and that (2) SDI is non-information-theoretic (what does that precisely mean?) and hence orthogonal to usual information theoretic loss in AT. Please clarify why this is not contradictory or resolve this.

	3. Include discussion on the computational overhead of the proposed method.

Beneficial but not critical:

	1. Provide a more rigorous theoretical analysis of the relationship between SDI and existing adversarial training objectives.

	2. Provide guidelines or heuristics for selecting the optimal β value for different scenarios.

	3. Explore the potential of SDI in other machine learning tasks beyond adversarial robustness.

**Strengths And Weaknesses:**

Strengths:
	1. Novel approach: The SDI measure provides a new perspective on adversarial robustness, complementing existing methods.
	2. Comprehensive empirical evaluation: The authors test their method across multiple datasets, model architectures, and attack types.
	3. Improved performance: Consistent improvements in robustness are demonstrated, especially against attacks like CW and Auto-attack.
	4. Generalization: The method shows better generalization to unseen attacks compared to baseline methods.
	5. Compatibility: The SDI regularization can be easily integrated with existing adversarial training techniques.
Weaknesses:
	1. Lack of theoretical guarantees: While the empirical results are strong, the paper lacks rigorous mathematical proofs for the conceptual similarities claimed.
	2. Computational cost: The paper doesn't thoroughly discuss the additional computational overhead of the SDI regularization.
Hyperparameter sensitivity: The impact of the β hyperparameter is shown, but guidelines for optimal selection are not provided.

---

> ### Author Response · Authors · 2024-10-11
>
> *We thank the reviewer for the valuable feedback. We have addressed the requested changes in the revision. For easy reference, we have used blue color to mark major changes in the text of the revised paper.  *
>
>
> **"Does the argument on alignment of SDI measure with minimization and maximization hold only when the maximum probability class is the true class? Please clarify."**
>
> Yes, the argument holds when the class with the highest probability corresponds to the true class. In this case, maximizing the SDI measure further widens the probability gap between the correct class and the incorrect ones, which enhances the model's learning by reinforcing the correct classification.
>
> **"Two of the main claims seems to be slightly contradictory: (1) you argue that SDI measure minimization and maximization aligns with minimization and maximization of AT and that (2) SDI is non-information-theoretic (what does that precisely mean?) and hence orthogonal to usual information theoretic loss in AT. Please clarify why this is not contradictory or resolve this."**
>
> The SDI metric minimization and maximization, and the non-information-theoretic measures we mentioned are totally independent of each other. Therefore, there should be no contradiction. The argument made in Sec. 3.2 about optimizing  the SDI measure aims to make a connection between the conventional inner maximization (finding adversarial examples) and outer minimizing (training or improving training) of AT and the  SDI minimization (finding adversarial examples) and SDI maximization (improving training), respectively.
>
> We use information-theoretic losses to refer to the family of  losses based on information theory concepts like Shannon entropy, cross-entropy, KL-divergence, etc.  In related works, adversarial examples are often crafted using variants of these losses, especially the cross-entropy and KL-divergence losses. Similarly, many losses and regularization terms are based on variants of these losses. Our argument is that the proposed SDI measure does not belong to this family. Instead, it is extended from the popular standard deviation measure, yet it can find good adversarial examples (See Algorithm 2 and Sec. 4.6).
>
>
> **"Include discussion on the computational overhead of the proposed method."**
>
>       We have added Section 4.5 to discuss the computational overhead.
>
>
> **"Provide guidelines or heuristics for selecting the optimal β value for different scenarios."**
>
> The optimal β value is tuned using a validation set, and the value of  β achieving a good balance between natural accuracy and robustness was selected. Note that we have improved the presentation of how β was selected in the revised paper (See Sec 4.4.4).
>
> **"Explore the potential of SDI in other machine learning tasks beyond adversarial robustness."**
>
> While the scope of our research is limited to improving adversarial robustness, we believe that the proposed regularization can be potentially applied in other supervised learning settings with the goal of improving learning.

---

### Review · Reviewer_K7kB · 2024-09-25

**Summary Of Contributions:**

This paper introduces a standard deviation-inspired (SDI) regularization term for adversarial training (AT) to improve adversarial robustness and generalization. Inspired by the concept of the standard deviation that quantifies the data points' dispersion, the authors use the standard deviation as a regularizer at the outer loop. Experimental results demonstrate that combining SDI with existing AT variants strengthens defenses against advanced attacks like CW and AutoAttack, while also improving robust generalization. The paper offers compelling improvements based on the standard AT approaches like TRADES.

**Audience:**

Yes

**Broader Impact Concerns:**

This paper does not appear to present any ethical concerns.

**Claims And Evidence:**

Yes

**Requested Changes:**

See the above comments.

**Strengths And Weaknesses:**

Pros)
+ This paper is well-written and relatively easy to follow.
+ A straightforward metric, known as the standard deviation-inspired (SDI) measure, could lead to performance improvements across various datasets and threat models.

Cons)
- Lack of intuition and insight into why SDI would be effective, and I remain unconvinced that SDI brings significant advantages to adversarial training. Could the authors provide more detailed explanations or insights to support this?
- The notations are slightly confusing. Eq.(5) utilizes $M_{SDI}$ as the inner loop for the formulation of adversarial examples, but Algorithm 1 presents instead using $L_{CE}$ (at line 7). This stems from confusion between the proposed formulation in Section 3.2 and the regularization introduced in Section 3.3.
- The experimental results are somewhat limited. While ResNet-18 and WideResNet-34-10 are commonly used backbones in the field for comparison, relying solely on them restricts the analysis. I believe that evaluating the proposed method with larger or more lightweight backbones could provide additional insights and would demonstrate further benefits over other approaches.
- (Minor) Some "inner" and "outer"s are misused (likely due to mistakes), but they should not be.
- This paper could be stronger if it includes comparisons with existing adversarial training methods [1, 2, 3, 4], as there is currently limited comparison with competing approaches.

  [1] LAS-AT: Adversarial Training with Learnable Attack Strategy, CVPR 2022

  [2] CAT: Collaborative Adversarial Training, arxiv 2023

  [3] Randomized Adversarial Training via Taylor Expansion, CVPR 2023

  [4] Boosting Adversarial Training via Fisher-Rao Norm-based Regularization, CVPR 2024

Further comments)
- I am still curious how such a simple metric would function as a regularizer with universality. If the data modality is not singular (I mean, there could be many clusters in each class), can the authors guarantee the effectiveness of the proposed metric?
- Could you elaborate on why the multi-class margin term should be included in the formulation in eq.(6)?
- Can the authors provide more results with other backbones over WideResNet, such as ResNet-18, and datasets like Tiny-ImageNet in Table 6?

---

### Decision · Action_Editor_vPS9 · 2024-12-01

**Recommendation:** Accept with minor revision

**Comment:**

The paper is easy to follow, the proposed method is simple, and experimental results show the benefits of the proposed method. All 3 reviewers recommend acceptance. Reviewers asked for clarification of the intuition and motivation of the SDI measure, experiments (certain baselines such as https://arxiv.org/abs/2303.10653 https://arxiv.org/abs/2403.17520, and computational comparisons, hyper-parameter selection), and also discussions about similar regularization-based approaches such as https://arxiv.org/abs/2010.02558 and https://arxiv.org/abs/1906.03749. The authors have provided a response to each one along with additional experimental results, and most of them have been incorporated into the updated version of the paper. I request the authors to further include the discussions about similar regularization-based approaches, which was provided during the discussion phase.

Some other suggestions: There is a period in the name of section 2.2 which can be removed. $\boldsymbol{\mathrm{N}}$ (after eq3) and $\boldsymbol{\mathrm{L}}$ (after eq4) seems different from the font within the equation. I would be more comfortable if the authors do not remove the standard deviations in the experimental results, and would be better if it is added back to the tables. In Table 8, bold number for each column is missing. One of the reviewer suggests the following in the final recommendation (which I believe is not visible to the authors): "However, my remaining concern is that the intuition can be still presented more straightforwardly. For example, though the authors explain "why SDI would be effective" in Sections 3.2 and 3.3, I believe conveying it as a simple and clear message could draw more attention to this paper."

**Audience:**

Adversarial robustness is a large research area within the machine learning community. I believe the audience of TMLR (especially researchers working in related fields) would be interested in the findings of this paper.

**Claims And Evidence:**

This paper proposes a regularization term based on a standard deviation-inspired (SDI) measure to improve adversarial robustness in multi-class classification problems. The SDI measure is an instance-level measure that captures how spread out the estimated class probabilities are relative to the class probability that is specified by the supervised label of the sample, e.g., a smaller value of this would imply more evenly distributed values. This is basically the regularization term that is used, except that the regularizer would become zero when the multi-class margin, i.e., estimated probability for the labeled class is smaller than the largest value of other classes, is negative. The proposed method is supported by experimental results, which show comparisons with various baseline methods and threat models. The paper also discusses sensitivity of the hyper-parameter (strength of the regularization term), computational cost, and the benefit of using the SDI metric for crafting adversarial examples.